# Suppressing Inflammation for the Treatment of Diabetic Retinopathy and Age-Related Macular Degeneration: Dazdotuftide as a Potential New Multitarget Therapeutic Candidate

**DOI:** 10.3390/biomedicines11061562

**Published:** 2023-05-27

**Authors:** Brice Nguedia Vofo, Itay Chowers

**Affiliations:** Department of Ophthalmology, Hadassah—Hebrew University Medical Center, Jerusalem 91120, Israel; vofo@hadassah.org.il

**Keywords:** diabetic retinopathy, diabetic macula edema, age related macular degeneration, anti-inflammatory compounds, anti-vascular endothelial growth factor, dazdotuftide

## Abstract

Diabetic retinopathy (DR) and age-related macular degeneration (AMD) are major causes of blindness globally. The primary treatment option for DME and neovascular AMD (nAMD) is anti-vascular endothelial growth factor (VEGF) compounds, but this treatment modality often yields insufficient results, and monthly injections can place a burden on the health system and patients. Although various inflammatory pathways and mediators have been recognized as key players in the development of DR and AMD, there are limited treatment options targeting these pathways. Molecular pathways that are interlinked, or triggers of multiple inflammatory pathways, could be promising targets for drug development. This review focuses on the role of inflammation in the pathogenesis of DME and AMD and presents current anti-inflammatory compounds, as well as a potential multitarget anti-inflammatory compound (dazdotuftide) that could be a candidate treatment option for the management of DME and AMD.

## 1. Introduction

Diabetic retinopathy (DR), and age-related macular degeneration (AMD) are the major causes of blindness in the developed world [1]. The international diabetes federation estimates that the global population with diabetes mellitus will approximately double between 2019 and 2045 [2]. The global prevalence of DR and clinically significant diabetic macular edema (DME) amongst diabetic patients is estimated to be 22.27% and 4.07%, respectively [3]. In DME, macular thickening and exudation of lipids alter the macular structure and are associated with vision loss. AMD is the leading cause of central vision loss in people age over 50 years in developed countries [4]. Since age is a major risk factor for AMD, with a rapidly aging population and increasing lifespan, the prevalence of AMD is also expected to rise [5]. In 2020, it was estimated that 196 million people live with AMD, and these numbers were projected to rise to 288 million by 2040. AMD accounts for 8.7% of all blindness worldwide [5,6]. It is classified into non-neovascular/atrophic or “dry” AMD (aAMD) and neovascular AMD (nAMD). In aAMD, atrophy is defined as complete retinal pigment epithelium and outer retina atrophy, which is referred to as geographic atrophy once it becomes confluent. In nAMD, macular neovascularization develops, leading to exudation and subsequent scaring and atrophy.

Anti-vascular endothelial growth factor (VEGF) compounds are the main treatment modality used for DME and nAMD, while suppressing complement activation is the only available treatment modality for aAMD. While anti-VEGF therapy proved to be efficient for DME and nAMD, insufficient response is common in these conditions [7,8]. In aAMD, available complement inhibitors achieve only modest inhibition of atrophy growth. Thus, an important need for additional more effective therapies exists for these conditions. Inflammation is a major pathway involved in the pathogenesis of DME and both forms of AMD. Here, we describe the rational for anti-inflammatory treatment in these conditions as well as available and future potential treatments.

## 2. Inflammation in the Pathogenesis of DME

Hyperglycemia in diabetic patient triggers several biochemical processes leading to inflammation, ischemia, and a pro-angiogenic state in the retina, with subsequent complications of vascular leakage, edema, neovascularization, and neurodegeneration [9,10]. Several pathways have been involved in the process.

### 2.1. Toll-Like Receptor Activation

Hyperglycemia enhances the expression and activation of Toll-like receptor 4 (TLR4) in human endothelial cells that may play an important role in DR. TLR4 is a pattern recognition receptor normally found in different types of cells in the retina, such as RPE, photoreceptors, microglial cells, astrocytes, Müller cells, and retinal vascular endothelial cells [11,12]. Hyperglycemia also upregulates endogenous TLR4 ligands, including the high-mobility group box 1 (HMGB1) [13,14,15]. TLR4 activation contributes to increased leukostasis through myeloid differentiation factor 88 (MyD88)-dependent pathways in leukocytes, with subsequent adhesion to retinal vessel walls [16]. Such adhesion triggers further inflammation, leading to endothelial cell death, pericyte loss, and vascular occlusion; all of which will lead to ischemia, hypoxia, and neovascularization [16,17]. The HMGB1-TLR4 signaling cascade also leads to translocation of nuclear factor kappa-light-chain enhancer of activated B cells (NF-κB) into the nucleus, resulting in the release of pro-inflammatory and pro-angiogenic cytokines, such as VEGF, basic fibroblast growth factor (bFGF), tumor necrosis factor (TNF)-α, interleukin (IL)-1, IL-6, and IL-8 [18]. Upregulation of these cytokines has multiple pro-inflammatory consequences including macrophage and endothelial cell activation. Upregulation of TLR4 has also been associated with increased levels of biomarkers of oxidative stress such as malondialdehyde, over-production of reactive oxygen species (ROS), and diminished antioxidant activities of super-oxide dismutase, catalase, and glutathione peroxidase [19,20,21]. TLR4 is also involved in glia (astrocyte and Müller cells) activation, contributing to neovascularization in hypoxic retina [22,23,24]. These glial cells have been shown to secrete VEGF [25]. Interestingly, TLR4 and its ligand HMGB1, are in a positive feedback loop, possibly perpetuating a chronic inflammatory and angiogenic state in the retina [26].

### 2.2. Polyol Pathway and Advanced Glycation End Products

Excess glucose is metabolized by the polyol pathway through aldose reductase which produces sorbitol. Sorbitol is impermeable to the cellular membrane, it accumulates in cells and induces osmotic damage [27,28]. Sorbitol can be metabolized into fructose-3-phosphate and deoxyglucosone via the sorbitol dehydrogenase-mediated pathway. These bi-products are glycolyzing agents that will lead to the deposition of advanced glycation end products (AGEs) [29,30,31]. The biochemical process involved in the sorbitol dehydrogenase-mediated pathway, and the upregulation of the polyol pathway, result in increased ROS and oxidative stress. Accumulated AGEs might crosslink proteins, altering their function, and affecting blood vessel wall components, the basement membrane, and cellular receptors [30]. AGEs could further induce damage by activating the cognate receptors to induce pro-inflammatory and pro-oxidant events that will drive oxidative stress and leukocyte adhesion in DR [30], leading to pericyte and capillary loss, and microaneurysm formation [32,33].

### 2.3. Protein Kinase C Pathway

The increased flux of glucose via the glycolytic pathway leads to elevated diacylglycerol levels that activate the protein kinase C (PKC). Apart from the plethora of biochemical processes triggered by this pathway that drives overexpression of nicotinamide adenine dinucleotide phosphate oxidase and NFkB in a number of vascular cells, exacerbating oxidative stress and inflammatory processes [34], the PKC-β isoform also drives VEGF expression [34].

### 2.4. RAAS System

The expression of the angiotensin-converting enzyme (ACE) in the retina has been reported to adversely affect capillary perfusion and vascular structure, and mediate VEGF upregulation. The accumulation of glucose and its metabolite, succinate, has been shown to activate the renin–angiotensin–aldosterone system in DR eyes [35]. Treatment with fosinopril (an ACE inhibitor) was found to improve the pathological and biochemical markers of DR in streptozotocin-induced diabetes in rats. Moreover, the upregulation of ACE in their serum and TGF-β1 in their pathological outer and inner nuclear layers of the retina were reduced. Hence, ACE-mediated TGF-β1 activation seem to play a role in the destruction of the blood-retina barrier during DR [36]. Moreover, high ACE concentrations were recently found in the blood serum of both DME and PDR patients by Neroev et al. [37].

### 2.5. Macrophage Polarization

Macrophages can progressively aggravate an inflammatory state under hyperglycemic conditions. Sofia et al. found that under long-term hyperglycemic treatment, macrophages secreted excessive inflammatory mediators while their phagocytosis and bactericidal functions were damaged at the same time [38]. Castro et al. also showed that macrophages exhibited a pro-inflammatory M1 phenotype after being exposed to high glucose both in vitro and in patients with hyperglycemia [39]. Glucose transporter 1-mediated glucose uptake in macrophages promotes glycolysis and ROS production, further inducing a pro-inflammatory phenotype of macrophages [40]. In mouse peritoneal macrophages, exposure to a high concentration of glucose (25 mM d-glucose) resulted in the increased levels of mRNA transcription of IL-1b, TNF-α, IL-6, IL-12, and iNOS through JUN N-terminal kinase/NF-κB [41].

### 2.6. Oxidative Stress

Oxidative stress mediated by the production of superoxide has been proposed as the unifying mechanism that links all the hyperglycemia-induced biochemical and molecular pathways discussed above [42,43]. Parallel to its ability to induce VEGF expression, there is evidence that oxidative stress can increase NF-κB, resulting in upregulation of genes involved in the immune and inflammatory processes as well as cellular proliferation and apoptosis; upregulate adhesion molecules (ICAM-1, VCAM-1, Integrins, and selectins); activate microglial cells to secrete inflammatory mediators (IL-1, IL-2, IL-6, IL-8, TNF-α); and increase expression of Monocyte Chemoattractant Protein 1 and Macrophage Inflammatory Protein 1 α gene [44].

Inflammation and leukostasis, resulting from the above mentioned mechanisms, induce capillary occlusion and hypoxia, which further stimulate VEGF expression and the accompanying vascular abnormalities that characterize DR [45]. Indeed, the use of various anti-inflammatory drugs, such as Nepafenac and triamcinolone, have all been shown to reduce VEGF expression, diminish leukostasis, reduce vascular permeability, inhibit retinal cell death, and improve VA [46,47,48]. Although VEGF expression and breakdown of the blood–retinal barrier has been well documented as a cause of DR and DME, there are other pathways through which retinal inflammatory mediators can affect vascular hyperpermeability and leakage, such as TLR4, interleukin-1β, and TNFα [11,49,50].

## 3. Inflammation in the Pathogenesis of AMD

### 3.1. Complement Activation

Inflammation is a major pathway in the pathogenesis of aAMD and nAMD. Several lines of evidence implicate the complement cascade and mononuclear cells in AMD. Components of the complement system were identified in AMD eyes, particularly in the retinal pigment epithelium (RPE) and drusen, the hallmark lesions of the disease. Systemic activation of the complement cascade was also detected, and genetic variants in complement genes which lead to accelerated complement activation are associated with increased risk for developing AMD [51,52,53]. The excessive activation of complement can lead to RPE and photoreceptor cell death, as well as chronic inflammation, oxidative stress, and angiogenesis, which ultimately contribute to vision loss [54,55]. Thus, targeting the complement cascade has emerged as a promising therapeutic approach for AMD, with several drug and gene therapy strategies currently in development that aim to reduce complement activation in AMD. Recently, the first compound based on complement inhibition (pegcetacoplan) was approved by the FDA for the treatment of aAMD. Monthly or bi-monthly intravitreal injections of this compound are associated with approximately 20% reduction of the progression of atrophy [56,57].

### 3.2. Mononuclear Cells

Mononuclear cells, such as macrophages and microglia, were also implicated in AMD. Macrophages accumulate in the vicinity of AMD lesions, where they can phagocytose debris and secrete pro-inflammatory and proangiogenic cytokines and chemokines. These cells can also interact with the complement system and modulate its activity, further exacerbating the immune response and tissue damage [58,59]. M1 macrophages are known to promote inflammation and angiogenesis, while M2 macrophages are associated with the resolution of inflammation and tissue repair [60]. Studies have suggested that a shift from M1 to M2 macrophages is anti-inflammatory. In vitro M2 macrophages have been shown to reduce the production of pro-inflammatory cytokines, such as IL-1β and TNF-α, and promote the production of anti-inflammatory cytokines, such as IL-10 and TGF-β, in response to RPE cells and choroidal endothelial cells [61,62]. In addition, M2 macrophages can inhibit the proliferation, migration, and tube formation of choroidal endothelial cells, which are critical for the development of CNV, the hallmark of nAMD [63]. In vivo, a shift from M1 to M2 macrophages also demonstrated anti-inflammatory properties, for example, the intravitreal injection of IL-4, a cytokine that promotes the activation of M2 macrophages, resulted in reduced CNV lesion size and vascular leakage in a mouse model of nAMD [64]. Similarly, the depletion of M1 macrophages or the polarization of macrophages towards an M2-like phenotype resulted in a reduction in CNV lesion size and vascular leakage in mouse models of nAMD [63,65]. Polarized macrophages can excrete cytotoxicity and/or increase abnormal neovascularization (CNV) and may thus be targeted to treat AMD [66,67,68,69,70,71].

### 3.3. TLR4

TLR was reported to be expressed in many cells, including RPE, photoreceptors, astrocytes, microglia, and retinal vascular endothelial cells [72]. RPE cells are activated to secrete pro-inflammatory and pro-angiogenic cytokines [73,74]. TLR activation in the retina induces cell death, and degenerating retina cells further stimulate TLR cells [75,76,77,78]. This leads to a vicious cycle of TLR activation, pro-inflammatory and angiogenic signaling, cell death, and further activation. In a systematic review by Klettner et al., the potential role of TLR activation of RPE in the development of AMD was summarized into the following: complement activation, pro-inflammation and pro-angiogenesis, neuronal degeneration, reduced RPE cell functions (barrier function, phagocytosis, function in visual cycle), and RPE cell degeneration leading to more TLR4 activation [79]. Amyloid-β, a component of drusen, can upregulate TLR4 and NF-κB expression and lead to progression of AMD [80,81]. TLR4 inhibition could attenuate expression of inflammatory and angiogenic factors, particularly IL-6, IL-8, IL-33, bFGF, and VEGF [82]. Moreover, a VEGF and CNV suppressive effect of calcium supplementation has been reported and associated to its ability to decrease the expression of TLR4, NF-κB, and Hif-1α in RPE cells [83].

### 3.4. Additional Factors

Other factors exacerbating inflammation in AMD are oxidative injury and smoking [84,85,86]. The components of drusen, A2E, oxidative stress, and VEGF-A expression, have been shown to activate NLRP3, leading to the activation of caspase-1 cascade [80,87,88,89]. Caspase-1 activates IL-18 and IL-1β [90]. IL-1β is a potent activator and mediator of inflammation, it is also a strong pro-angiogenic factor that drives VEGF production [91]. Il-18 has also been implicated in several inflammatory conditions [92]. Oxidative stress, and ROS, NLRP3, complement cascade, Ang2, macrophage recruitment and polarization, hypoxia of the RPE are all factors that promote the production of VEGF, angiogenesis, and the progression in nAMD. VEGF-A strongly induces vascular proliferation and migration of the endothelial cells essential for both physiological and pathological angiogenesis [93]. Currently the mainstay of treatment for nAMD is anti-VEGF therapy. However, more efforts in the development of therapies targeting the upstream triggers of VEGF and additional pro-angiogenic factors could complement and supplement current therapies for nAMD.

## 4. Anti-Inflammatory Compounds for the Treatment of DME and AMD

### 4.1. Anti-Inflammatory Effect of Anti-VEGF Compounds

Intravitreal injection of anti-VEGF compounds is the standard of care treatment of DME. Pivotal studies have demonstrated the efficacy and safety of several biologic anti VEGF compounds among them aflibercept [7], ranibizumab [94], and faricimab [95], as well as more recently biosimilar compounds such as Formycon and Bioeq’s FYB201, MYL-1701, Razumab, ranibizumab-nuna, and ranibizumab-eqrn [96,97,98]. Interestingly, many studies have demonstrated that anti-VEGF compounds were not only anti-angiogenic, but also possess anti-inflammatory properties which may contribute in part in their success in the management of retinal pathologies [99,100,101]. By inhibiting VEGF, anti-VEGF compounds can reduce the recruitment of immune cells and the production of pro-inflammatory cytokines, thereby exerting their anti-inflammatory effects [102]. Additionally, VEGF has been shown to enhance the permeability of blood vessels, which can contribute to the accumulation of inflammatory cells and fluid in tissues. By inhibiting VEGF, anti-VEGF compounds can reduce vascular permeability and thereby limit the accumulation of inflammatory cells and fluid in tissues [103].

### 4.2. Corticosteroids

Corticosteroids have long been recognized as a potential treatment option for DME. Dexamethasone, triamcinolone, and fluocinolone have been used in many forms, including particulate suspension, viscoelastic mixtures, and solid slow-release devices [46,104,105,106]. The enthusiasm was initially high using different dosages and treatment intervals [107]. However, while protocol B trial of the DRCR network proved that focal laser led to higher VA gains at three years compared to intravitreal triamcinolone injections, protocol I compared the effectiveness of intravitreal ranibizumab, intravitreal triamcinolone acetonide, and focal/grid laser photocoagulation for the treatment of diabetic macular edema, and it concluded that intravitreal ranibizumab and intravitreal triamcinolone acetonide were both superior to focal/grid laser photocoagulation for improving visual acuity in patients with DME. However, the study also found that the use of intravitreal triamcinolone acetonide was associated with a higher risk of complications, including increased intraocular pressure and cataract progression [47,108,109]. The protocol U of the DRCR network demonstrated that combining a dexamethasone implant with anti-VEGF therapy (ranibizumab) resulted in moderate gains in VA compared to continuous anti-VEGF treatment alone, in the short term [110]. A literature review of articles comparing dexamethasone implants to anti-VEGF therapy in DME also reported similar VA outcome in observational studies, and superior VA gains in the dexamethasone implant group, in real life studies, although the authors argue that these differences could be due to differences in the baseline VA and the number of anti-VEGF injections in different studies [111]. However, according to the findings reported in a recent meta-analysis, intravitreal steroid treatment for DME was associated with no significant difference in BCVA compared to anti-VEGF treated eyes, although with a significantly lower retinal thickness [112]. Corticosteroid therapy has also been shown to be potentially useful in DME eyes that respond poorly to anti-VEGF [113,114,115].

Intravitreal injections of triamcinolone (as off-label use) were demonstrated to yield a two-step reversal in the DR severity score compared to focal/grid laser at three years [116]. The DR-Pro-DEX study and others also indicated that dexamethasone and fluocinolone implants significantly delayed the progression and reduced the severity of DR [117,118]. The anti-inflammatory effect of steroids is thought to be due to their ability to reduce the levels of several pro-inflammatory cytokines, including IL-1β, IL-6, IL-8, tumor TNF-α, and VEGF in the vitreous of patients with DR [104,119]. In addition to reducing cytokine levels, steroids have also been shown to modulate several other molecular pathways involved in the pathogenesis of DR, including the regulation of extracellular matrix proteins, the suppression of ROS production, and the modulation of cellular apoptosis and survival pathways [120,121].

The side effects of corticosteroids remain the main limitation to their use for the management of these ocular pathologies. Cataract in phakic eyes and intraocular pressure elevation regardless of lens status has been reported in varying degrees in all studies [122,123,124,125,126,127]. Studies indicate that up to 32% of eyes treated with corticosteroids may develop high (≥25 mmHg) intraocular pressure [127,128,129]. In addition, steroid treatments induce lens opacification. The Score study [5] estimated the 12 month incidence of new onset lens opacities or progression of lens opacities to be up to 33%, and by 24 months, over 33% of patients required cataract surgery [130]. In addition, there are other less frequent complications of steroid use, such as non-infectious endophthalmitis, and pseudo-endophthalmitis, activation of ocular and peri-ocular infections, and steroid-induced central serous chorioretinopathy [131].

### 4.3. Rho-Associated Protein Kinase Inhibition

Ripasudil is a Rho-associated kinase inhibitor developed originally for the treatment of glaucoma and ocular hypertension. Rho-associated protein kinase (ROCK) 1 and 2 determine macrophage polarization into M1 and M2 subtypes. Aging is known to increase ROCK2 signaling, thereby leading to the over-expression of proangiogenic macrophages associated with increased IL-4 production and angiogenesis. A recent study showed that NLRP3, apoptosis-associated speck-like proteins containing a CARD (ASC), caspase1, IL-1β, and IL-18 were inhibited by ripasudil [132].

### 4.4. None-Steroidal Anti-Inflammatory Drugs

NSAIDS are potent inhibitors of cyclooxygenase enzymes and therefore suppress the synthesis of pro-inflammatory prostaglandins. Studies have also described the role of pro-inflammatory prostaglandins in the pathogenesis of DR and AMD, with recent studies searching for the therapeutic role of NSAIDs for these disorders [133]. Moreover, NSAIDS have been detected and measured in the vitreous cavity after topical application, making them more attractive as a convenient route of intravitreal drug administration [134].

In AMD, the use of NSAIDS has been mentioned in the management of nAMD, but only in combination with anti-VEGF. For example, in 2015, a pilot study described higher efficacy in terms of visual acuity improvement when a topical NSAID (bromfenac) was associated to aflibercept injection (EYLEA^®^, 2 mg, Regeneron Pharmaceuticals, Inc., New York, NY, USA) compared with a single anti-VEGF therapy (EYLEA^®^, 2 mg), although the anatomical outcome was the same. Cox-2 has been detected in human CNV membranes and there is scientific evidence that it is a promoter of CNV [135,136]. Hence, the pharmacological inhibition of COX appears to reduce VEGF expression in mice models [137].

In DR, scientific evidence indicates that retina cells consistently upregulate COX and prostaglandins [138,139], hence the potential place of NSAIDS in their management. The therapeutic potential of NSAIDS was first suspected a century ago, when rheumatoid arthritis patients on salicylates had reduced incidence of DR [140], since then, several studies have examined the clinical benefit of NSAIDS given both systemically [141], topically [142], and even intravitreally [143]. However, most of these studies are small, retrospective, uncontrolled studies [144]. Despite the considerable scientific rational, Protocol R of the DRCR network examined the effect of topical NSAIDS for non-central DME in a multi-center double-masked randomized trial, and concluded that, at one year, there was no meaningful effect on OCT measured retinal thickness [145]. Due consideration must also be given to the potential side effects associated with the long term use of NSAIDS, especially amongst patients whose corneas are compromised by diabetes, ocular surgery, or auto-immune disease. For example, although rare, corneal melting secondary to NSAIDS use is a potential complication in this patient population and could be sight threatening [146].

## 5. Dazdotuftide

Dazdotuftide is a novel, small synthetic molecule. It is a peptide conjugate comprised of tuftsin and phosphorylcholine that are covalently attached. The phosphorylcholine (PPC) moiety is based on a sequence from helminths’ secretory molecules which has been shown to be responsible for their immunoregulatory functions [147]. Tuftsin is a naturally occurring endogenous immunomodulatory tetra-peptide (Thr-Lys-Pro-Arg) produced in the spleen by enzymatic cleavage of the Fc-domain of the heavy chain of IgG. The peptide is coupled to diazotized 4-aminophenyphosphorylchloride to form an azo bond between the tuftsin and PC [148]. Extensive research has been done on Tuftsin and PPC separately, and the activities of each of them towards immune regulation. Dazdotuftide has been suggested to provide a strong synergistic effect, far surpassing the efficacy of the compounds given separately [149].

### 5.1. Dazdotuftide In Vivo and In Vitro Studies

Dazdotuftide and its components demonstrated in various autoimmune animal models and in vitro test systems its inhibitory effect on TLR-4 (inhibition of NfKb cascade), NRP-1 (inhibition of VEGF 165), and ACE2. In vivo studies have demonstrated Dazdotuftide to be effective in the treatment of inflammation in autoimmune animal models: rheumatoid arthritis [150], lupus nephritis [148], and colitis [151]. Dazdotuftide’ ability to prevent and treat the disease in these models was accompanied by reduction of pro-inflammatory cytokine levels and an increase of anti-inflammatory cytokine expression, as well as expansion of T and B regulatory cells. It is proposed that the dual functionality of dazdotuftide is due to its binding to different receptors via its two moieties: the PPC targeting TLR4 [152], and leading to NF-κB inhibition and suppression of inflammatory cytokine synthesis, and the tuftsin end of Dazdotuftide targeting neuropilin-1 [153], leading to inhibition of VEGF 165 activity [154] and macrophage shift towards M2 anti-inflammatory macrophages that secrete IL-10 and induce Treg activation [155].

In vitro, dazdotuftide was tested on specimens of peripheral blood mononuclear cells (PBMCs) and temporal artery biopsies (TABs) obtained from patients with giant cell arthritis and age-matched controls. The PBMCs were activated by CD3/CD28 beads and tests were conducted to depict inflammatory cytokine secession and IL-10 anti-inflammatory cytokines. Upon treatment with dazdotuftide, there was a decrease in the production of IL-1β, IL-2, IL-5, IL-6, IL-9, IL-12(p70), IL-13, IL-17A, IL-18, IL-21, IL-22, IL-23, IFNγ, TNFα, and GM-CSF by activated PBMCs, with negligible effect on cell viability. Likewise, in inflamed TABs, treatment with dazdotuftide reduced the production of IL-1β, IL-6, IL-13, IL-17A, and CD68 gene expression. The effects of Dazdotuftide were considered superior to dexamethasone as a standard of care due to its greater reduction of IL-2, IL-18, and IFNγ in CD3/CD28-activated PBMCs, and CD68 gene in inflamed TABs [156]. Moreover, recent analysis by molecular docking has demonstrated the strong binding and inhibitory ability of tuftsin on ACE2 and NRP1 [157].

### 5.2. Dazdotuftide Inhibition of TLR and NRP-1

TLR (inhibition of NfKb), NRP-1 (increase reactivity of VEGF 165), and ACE activation have been demonstrated to play key roles in the cascade of events that leads to inflammation and contributes to the progression of DR and nAMD. In a diabetic retina, hyperglycemia enhances the expression and activation of TLR4 in human endothelial cells that may play an important role in DR [11] Cells exposed to high glucose had an increased expression of downstream factors of TLR4 including myeloid differentiation factor 88, an inflammatory cytokine inducer, and NF-κB [158,159], which may lead to the secretion of more inflammatory cytokines like IL-1β [160]. TLR4 polymorphisms were associated with a higher prevalence of retinopathy, further supporting the role of this gene in DR [161,162]. In an oxygen-induced retinopathy model, the absence of TLR4 was seen to be associated with low glial activation and low expression of HMGB-1, the endogenous ligand for TLR4. The expression of HMGB-1 in ischemic retina was shown to promote the production of pro-inflammatory factors and also initiate TLR4-dependent responses that contribute to neovascularization [23]. In fact, TLR4 signaling was implicated in inflammation, leukostasis, angiogenesis, blood–retinal barrier breakdown, hypoxia, cellular apoptosis, neurodegeneration, and oxidative stress in a diabetic retina. This is, therefore, a potential therapeutic target for DR, that has stimulated research around several compounds hypothesized to inhibit this receptor [10]. ACE activation has been implicated in the pathogenesis of DR and DME, potentially causing a disruption of the blood–retinal barrier in diabetic eyes [35,36,37]. Tuftsin, a component of dazdotuftide, has a strong binding affinity to ACE2 [157]. This is therefore another pathological pathway to DR and DME that could be inhibited by dazdotuftide (see Figure 1).

In the context of nAMD, TLR inhibition could prevent complement activation, inhibiting RPE cell loss in aAMD and downplaying the RPE cell-driven BRB breakdown, leukocyte recruitment, VEGF production, angiogenesis, and exudation in nAMD [79]. TLR inhibition could also break the vicious cycle of retina cell death and degeneration that leads to pro-inflammatory and angiogenic signaling and further TLR activation. In both DR and nAMD, neovascularization is driven by VEGF, and this cytokine may be suppressed via Dazdotuftide activity [75,76,77,78].

NRP1 functions as a co-receptor for VEGF and its inhibition would diminish VEGF bioactivity [154]. For example, NRP1 has been found to be present, alongside VEGFR-2, in the endothelial and RPE cells in nAMD eyes demonstrating subfoveal CNV [163], and variations in the NRP1 gene were associated with treatment response to anti-VEGF therapy in nAMD [164].

## 6. Conclusions

While anti-VEGF compounds are effective and useful for the treatment of DR/DME and nAMD, many eyes experience an insufficient effect, and these compounds are ineffective for aAMD. Inflammation is a major pathway in the pathogenesis of DR/DME, aAMD, and nAMD. Corticosteroids are the only anti-inflammatory compounds approved for the treatment of DR/DME, and anti-complement biologics are the only approved therapy for aAMD. While these therapies are effective, both are associated with significant side effects and have limited efficacy. Dazdotuftide combines several functions including anti-VEGF, anti-TLR, and M1 to M2 macrophage polarization which could be an added advantage in the management of DME, DR, and nAMD. Dazdotuftide’s ability to inhibit NRP1, TLR, and activate anti-inflammatory macrophages makes this multi-target new drug a potential new therapeutic option for DR/DME and AMD. Targeting TLR, an upstream molecule implicated in the pathogenesis of these posterior ocular pathologies, may also yield better durability of the therapeutic effect, with a lower intra-vitreal injection frequency required than the current anti-VEGF therapies. Additionally, dazdotuftide may be formulated as a slow-release implant, thereby further reducing treatment burden and increasing patient compliance, which is a crucial factor for successful treatment in chronic conditions such as DR/DME and nAMD. Clinical trials should now test the hypothesis that this compound may be useful for these indications.

## Figures and Tables

**Figure 1 biomedicines-11-01562-f001:**
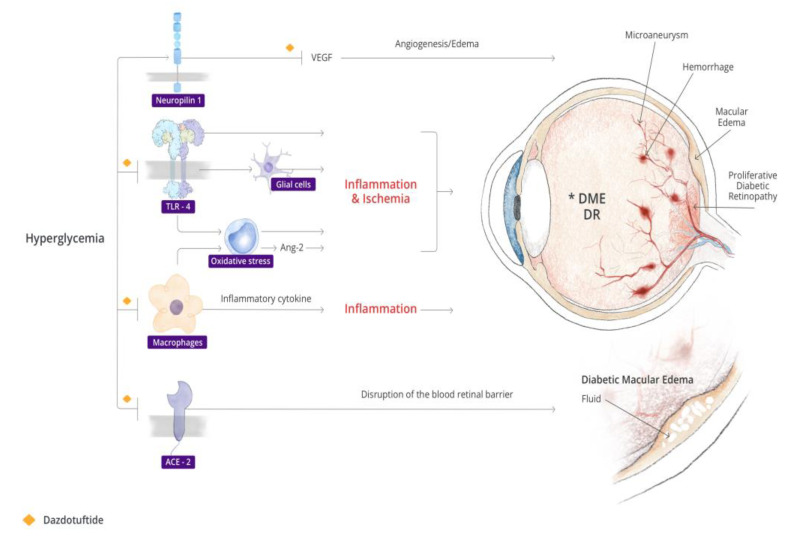
Illustration of potential sites for dazdotuftide inhibition along the pathogenic pathway of diabetic macula oedema and diabetic retinopathy (DR). * Diabetic macula oedema.

## Data Availability

No new data were created or analyzed in this study. Data sharing is not applicable to this article.

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
