# Peer review of "Suppressing Inflammation for the Treatment of Diabetic Retinopathy and Age-Related Macular Degeneration: Dazdotuftide as a Potential New Multitarget Therapeutic Candidate"

_biomedicines, 2023, doi:10.3390/biomedicines11061562_

Round 1

Reviewer 1 Report

1. line 244-45. It says: VEGF therapy in DME, also reported similar VA outcome in observational studies, and superior VA gains in the dexamethasone implant group, in real life studies.

 However in a recent meta-analysis: Overall, intravitreal steroid treatment for DME was associated with no significant differences inBCVA.

Patil NS, Mihalache A, Hatamnejad A, Popovic MM, Kertes PJ, Muni RH. Intravitreal Steroids Compared with Anti-VEGF Treatment for Diabetic Macular Edema: A Meta-Analysis. Ophthalmol Retina. 2023 Apr;7(4):289-299. doi: 10.1016/j.oret.2022.10.008. Epub 2022 Oct 19. PMID: 36272716.

2. Line 303: Consider adding: A risk of None-steroidal anti-inflammatory drugs induced corneal melt.

Rigas B, Huang W, Honkanen R. NSAID-induced corneal melt: Clinical importance, pathogenesis, and risk mitigation. Surv Ophthalmol. 2020 Jan-Feb;65(1):1-11. doi: 10.1016/j.survophthal.2019.07.001. Epub 2019 Jul 12. PMID: 31306671.

3. Line 358 and 389 correct : luekostasis, Dazdotuftidecombines

Author Response

Comment:

  1. line 244-45. It says: VEGF therapy in DME, also reported similar VA outcome in observational studies, and superior VA gains in the dexamethasone implant group, in real life studies.

 However in a recent meta-analysis: Overall, intravitreal steroid treatment for DME was associated with no significant differences inBCVA.

Patil NS, Mihalache A, Hatamnejad A, Popovic MM, Kertes PJ, Muni RH. Intravitreal Steroids Compared with Anti-VEGF Treatment for Diabetic Macular Edema: A Meta-Analysis. Ophthalmol Retina. 2023 Apr;7(4):289-299. doi: 10.1016/j.oret.2022.10.008. Epub 2022 Oct 19. PMID: 36272716.

Response

Thank you very much for pointing out this very interesting study. Indeed, it is a comprehensive meta-analysis that looked at the BCVA outcomes between anti-VEGF and Steroids treated eyes.

We have now modified this point, adding the following paragraph “However, according to the findings reported in a recent meta-analysis, intravitreal steroid treatment for DME, was associated with no significant difference in BCVA compared to anti-VEGF treated eyes, though with a significantly lower retinal thickness [107].” Line 250-255

We have also added the reference to this study. Thank you.

Comment:

  1. Line 303: Consider adding: A risk of None-steroidal anti-inflammatory drugs induced corneal melt.

Rigas B, Huang W, Honkanen R. NSAID-induced corneal melt: Clinical importance, pathogenesis, and risk mitigation. Surv Ophthalmol. 2020 Jan-Feb;65(1):1-11. doi: 10.1016/j.survophthal.2019.07.001. Epub 2019 Jul 12. PMID: 31306671.

Response:

Thank you very much for this suggestion. Indeed, this is a potential complication that could easily be overlooked. We have now included a sentence mentioning this.

Due consideration must also be given to the potential side effects associated with the long-term use of NSAIDS especially amongst patients whose corneas are compromised by diabetes, ocular surgery, or auto-immune disease. For example, though rare, corneal melting secondary to NSAIDS use is a potential complication in this patient population and could be sight threatening.” Line 310-315

Comment:

  1. Line 358 and 389 correct : luekostasis, Dazdotuftidecombines

Response:

We apologize for this inaccuracy. We have now corrected as follows; Leukostais, and Dazdotuftide combines, respectively.

Reviewer 2 Report

This review summarizes the molecular pathways of diabetic retinopathy and
age-related macular degeneration in the prospect of therapeutic applications targeting immune response. The two diseases have different etiologies, high glucose levels in blood vessels for the first and soft drusen bodies (fatty proteins accumulation and extracellular debris under the retina) that may cause bleeding for the second. For the authors of this review in both diseases the immune response underlies the problem therefore treatment of the inflammation is beneficial for the patients. The authors present anti-inflammatory compounds for the treatment of these diseases, in particular anti-VEGF, corticosteroids, Rho-associated protein kinase inhibitors, non-steroidal anti-inflammatory drugs, and dazdotuftide. Some of these compounds have been used in other pathological conditions related with inflammation as well. In summary, this is a significant contribution
with a collective approach on common treatments of diabetic retinopathy and age-related macular degeneration with particular interest for the readers
of biomedicines.

Author Response

Comment

This review summarizes the molecular pathways of diabetic retinopathy and
age-related macular degeneration in the prospect of therapeutic applications targeting immune response. The two diseases have different etiologies, high glucose levels in blood vessels for the first and soft drusen bodies (fatty proteins accumulation and extracellular debris under the retina) that may cause bleeding for the second. For the authors of this review in both diseases the immune response underlies the problem therefore treatment of the inflammation is beneficial for the patients. The authors present anti-inflammatory compounds for the treatment of these diseases, in particular anti-VEGF, corticosteroids, Rho-associated protein kinase inhibitors, non-steroidal anti-inflammatory drugs, and dazdotuftide. Some of these compounds have been used in other pathological conditions related with inflammation as well. In summary, this is a significant contribution
with a collective approach on common treatments of diabetic retinopathy and age-related macular degeneration with particular interest for the readers
of biomedicines.

Response:

Thank you very much for your appreciation of this work.

Reviewer 3 Report

This is a correct, broad perspective review on the the possible inflammatory background of two high-prevalence eye diseases, DME and AMD and their treatment options.

It would be worth mentioning in the title that a new therapeutic option still under testing will also be discussed.

L249 The intravitreal injection of triamcinolon is off label. The summary of product characteristics clearly indicates that the manufacturers do not recommend administration by intravitreal injection

Author Response

Comment:

This is a correct, broad perspective review on the possible inflammatory background of two high-prevalence eye diseases, DME and AMD and their treatment options.

It would be worth mentioning in the title that a new therapeutic option still under testing will also be discussed.

Response:

Thank you very much for this suggestion, we have now modified the title to read as follows:

Suppressing inflammation for the treatment of diabetic retinopathy and age-related macular degeneration: Dazdotuftide as a potential new multitarget therapeutic candidate.

Comment:

L249 The intravitreal injection of triamcinolon is off label. The summary of product characteristics clearly indicates that the manufacturers do not recommend administration by intravitreal injection.

Response:

Thank you very much for precising this, we have now edited to mention this to our readers. That statement now reads:

“Intravitreal injections of triamcinolone (as off-label use) were demonstrated to yield a 2-step reversal in the diabetic retinopathy severity score compared to focal/grid laser at 3 years” Line 255-257

Thank you very much.